# CORAL: CORRESPONDENCE ALIGNMENT FOR IMPROVED VIRTUAL TRY-ON

## ABSTRACT

Virtual Try-On (VTON) aims to outfit a person with a specific garment from paired person and garment images. Recent diffusion-based approaches show promising results but still struggle to preserve fine-grained details such as logos, patterns, and textures. We suggest these failures come from inaccurate query–key matching in attention maps. To analyze this, we introduce a correspondence evaluation framework that extracts dense correspondences from attention maps and evaluates them with pseudo ground-truth matches. Using this framework, we analyze a simple DiT-based baseline and observe that its attention maps in most layers fail to capture reliable semantic correspondences. We then propose **CORAL**, a lightweight regularization strategy with two components: correspondence loss, which corrects where each query attends by aligning it with reliable external matches, and entropy loss, which sharpens attention for more confident matching. CORAL improves person–garment alignment in our baseline and can be applied to other diffusion-based pipelines without architectural changes.

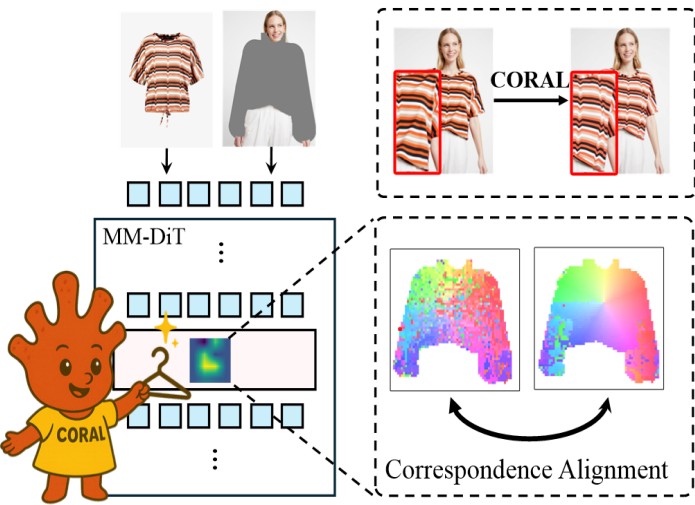

Figure 1: **Teaser. CORAL** aligns attention maps with reliable correspondences, resulting in better garment–person alignment and finer detail preservation in Virtual Try-On.

## 1 INTRODUCTION

Given a pair of person and garment images, Virtual Try-On (VTON) aims to synthesize the same person image wearing the given garment, accurately aligning the person and garment under large geometric variations. Recent advances in diffusion models have shown remarkable progress in VTON, driving growing applications in e-commerce and AR/VR. Conventional works often focus on preserving the fine-grained details of a given garment, through advanced inference techniques (Bhunia et al., 2023), additional conditioning signals (Choi et al., 2024; Kim et al., 2025), or garment encoders (Kim et al., 2023; Choi et al., 2024; Zhou et al., 2024).

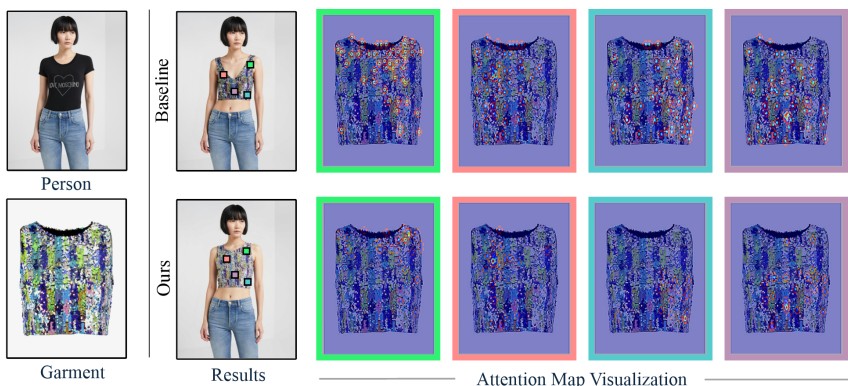

Figure 2: **Attention Map Visualization.** Compared to the baseline, our method produces sharper and more localized attention, resulting in more accurate person–garment correspondences.

However, they still fail to accurately preserve garment details, such as small logos or text, repetitive patterns, and textures. As discussed in (Zhou et al., 2024; Nam et al., 2024a; 2025a), this failure largely stems from imprecise *correspondence* estimation between the person's query and the garment's key in the attention module of diffusion models.

Recent works (Zhou et al., 2024) address this by improving correspondence-awareness of the model. This is achieved by applying an RGB photometric loss between the person image and the garment image warped by the dense flow map estimated from the attention module of the diffusion U-Net Zhou et al. (2024). However, the model learned through such supervision still struggles with repetitive patterns, homogeneous textures, or illumination changes, where the photometric assumption often fails because non-matching pixels may share similar RGB values (Truong et al., 2021). Inaddition, these methods focus solely on the diffusion U-Net, instead of advanced diffusion models such as Diffusion Transformers (DiTs), without analyzing how person–garment correspondences are represented within the model or how they could be further enhanced.

To address this, we first explore person–garment *correspondence* within Diffusion Transformers (DiTs) for VTON. We focus on the multi-modal attention mechanism in DiTs, where the person, garment, and text prompt interact. As highlighted in Figure 2, our analysis shows that the internal query–key matching between person and garment tokens is often weak, leading to misaligned and suboptimal try-on results.

Based on this analysis, we propose a new framework, **CORAL** (**COR**respondence **AL**ignment), which explicitly enhances query–key matching in the attention module of DiTs by aligning it with reliable correspondences obtained from the off-the-shelf vision foundation model DINOv3. Specifically, we introduce two complementary supervision signals: (1) **correspondence alignment loss**, which aligns the garment key points attended by each person query point with the correspondences provided by DINOv3, and (2) **entropy minimization loss**, which minimizes the entropy of the attention distribution to keep the attention sharp and localized.

In our experiments, we demonstrate that CORAL significantly improves query–key matching between the person and garment (Figure 2), thereby improving the VTON performance, and surpasses prior RGB photometric loss. Ablation studies further validate our design choices.

In summary, our main contributions are as follows:

- We provide new insights into VTON, demonstrating that accurate person–garment alignment depends on precise query–key matching within the attention module of DiTs.
- We introduce a novel VTON method that improves query–key matching through correspondence alignment with DINOv3 and entropy minimization, yielding sharper and more reliable matches.

## 2 RELATED WORK

**Image-based Virtual Try-On.** Early image-based virtual try-on methods (Han et al., 2018; Wang et al., 2018) commonly follow a two-stage framework: garment deformation and try-on synthesis.

Diffusion-based models have significantly advanced try-on quality. Some methods (Gou et al., 2023; Morelli et al., 2023) bridge GAN-based and diffusion-based approaches by incorporating traditional warping modules within the diffusion framework. Other methods (Choi et al., 2024; Kim et al., 2023; Nam et al., 2025b) employ ControlNet-like structures or an additional network to encode garment images. CATVTON(Chong et al., 2025) simply concatenates person and garment images along spatial dimensions as inputs, eliminating the need for additional encoding modules. Leffa (Zhou et al., 2024) proposes auxiliary regularization loss on attention maps to enforce proper alignment between target queries and reference keys. However, these approaches still struggle to preserve fine-grained details across diverse poses and garment types.

**Correspondence in Diffusion Models.** The development of diffusion models has inspired studies on their correspondence capabilities, revealing visual representations useful for cross-image understanding. DIFT (Tang et al., 2023) showed that diffusion features can establish semantic correspondences without extra training, and DiffMatch (Nam et al., 2024b) proposed a diffusion-based framework for dense matching using generative priors. Improving correspondence has also been shown to boost downstream tasks: DiffTrack (Nam et al., 2025a) explores temporal correspondences between video frames for better point tracking and video generation, while Track4gen (Jeong et al., 2025) supervises cross-frame correspondences to enhance video quality. These works mainly address feature-level correspondence. Other studies examine semantic correspondence in attention: Liu et al. (Liu et al., 2024) analyze cross- and self-attention to explain text–image alignment, and SynGen (Rassin et al., 2024) refines attention maps to improve text–object matching. Our work targets attention-level visual correspondence between images to improve generation quality.

## 3 PRELIMINARIES

**Diffusion Models.** The latent diffusion model (LDM) consists of a variational autoencoder (VAE) $\varepsilon$, a denoising network $\epsilon_\theta(\cdot)$. The VAE encoder $\varepsilon$ maps an image $x$ into a latent representation $z_0 = \varepsilon(x)$. At each diffusion timestep $t \in [0, 1]$, Gaussian noise $\epsilon \sim \mathcal{N}(\mathbf{0}, \mathbf{I})$ is added to $z_0$ according to a predefined noise schedule, yielding the noised latent $z_t = (1 - t)z_0 + t\epsilon$. Given conditioning tokens $\mathbf{c}$, the denoiser $\epsilon_\theta$ is trained to predict the added noise, with the objective

$$\mathcal{L}_{\text{diff}} = \mathbb{E}_{z_0, \epsilon, t} \left\| \epsilon - \epsilon_\theta(z_t, \mathbf{c}, t) \right\|_2^2. \tag{1}$$

Here, the conditioning tokens $\mathbf{c}$ can be text tokens from the text encoders or visual tokens extracted from reference images by a visual encoder.

**Multi-Modal Attention in DiTs.** In DiTs, attention is computed over a joint token sequence across the noisy latent tokens $z_t$ and the conditioning tokens $\mathbf{c}$. Here, conditioning tokens $\mathbf{c}$ are concatenated token-wise across multiple conditions, constructing $\mathbf{c} = [c_1; c_2; \ldots; c_{N_{\text{cond}}},]$, where $N_{\text{cond}}$ denotes the number of different conditions. The final input sequence can thus be written as $S = [z_t; \mathbf{c}]$. At each layer $l$ and head $h$ within the transformer blocks, $S$ is mapped into the queries and keys, formulated as

$$Q^{l,h} = [Q_{z_t}^{l,h}; Q_{c_1}^{l,h}; \ldots; Q_{c_{N_{\text{cond}}}}^{l,h}], \quad K^{l,h} = [K_{z_t}^{l,h}; K_{c_1}^{l,h}; \ldots; K_{c_{N_{\text{cond}}}}^{l,h}]. \tag{2}$$

To encode positional information, each token in $S$ is augmented with rotary position embeddings (RoPE). The attention map $A^{l,h}$ is then calculated by

$$A^{l,h} = \text{Softmax}\left(\frac{Q^{l,h}(K^{l,h})^\top}{\sqrt{d}}\right), \tag{3}$$

where $\text{Softmax}$ is applied over the key dimension for each query. This full attention encodes how each latent or conditioning token attends to the remaining tokens.

## 4 METHODOLOGY

### 4.1 TASK DEFINITION

Given a person image $I_p \in \mathbb{R}^{H \times W \times 3}$, a garment image $I_g \in \mathbb{R}^{H \times W \times 3}$, a pose condition $I_c \in \mathbb{R}^{H \times W \times 3}$, and binary masks $M_p, M_g \in \{0, 1\}^{H \times W \times 1}$ indicating the person and garment regions

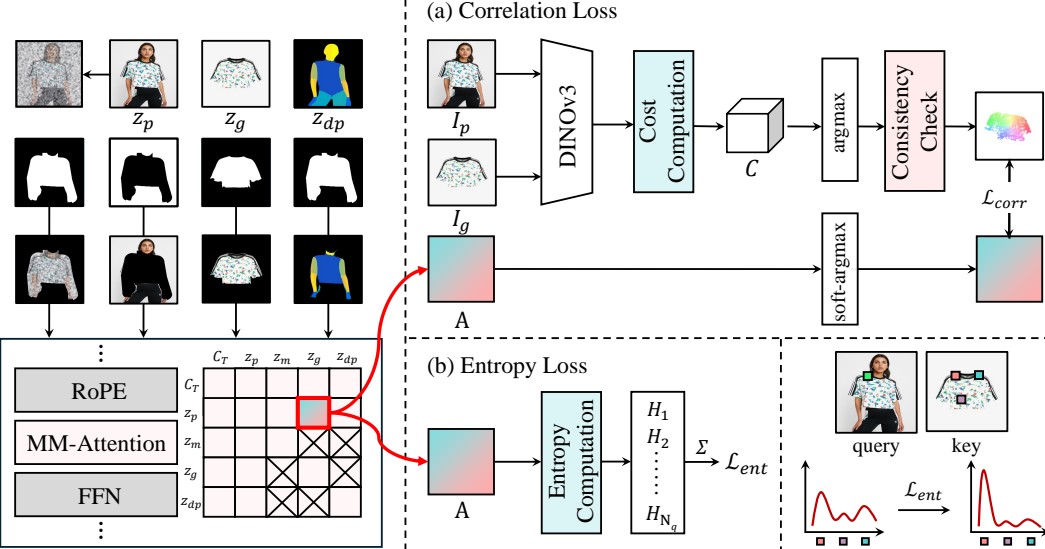

Figure 3: **Overall Architecture.** The person image $I_p$, garment image $I_g$, and pose condition $I_{dp}$ are encoded into latents $z_p$, $z_b$, $z_g$, and $z_{dp}$, which are processed by multi-modal attention layers in DiTs. We enhance query–key matching using a correspondence loss $\mathcal{L}_{\text{corr}}$ supervised by DINOv3 and an entropy loss $\mathcal{L}_{\text{ent}}$ that sharpens attention distributions, both applied to the attention with person tokens as queries and garment tokens as keys.

in $I_p$, our goal is to generate a try-on image $\hat{I}$ that preserves the identity of $I_p$ and the pose of $I_c$, obatined from Densepose (Güler et al., 2018), while replacing the clothing with the appearance of $I_g$. Note that $M_p$ denotes the entire upper-body region excluding hands, obtained by dilating the human-parsing contour mask, following prior work (Choi et al., 2024).

## 4.2 Diffusion Transformers (DiTs)-based Baseline

We encode the person image $I_p$, the garment image $I_g$, and the pose condition $I_c$ into the latent space using a VAE encoder $\varepsilon$, producing $\varepsilon(I_p)$, $\varepsilon(I_g)$, $\varepsilon(I_c) \in \mathbb{R}^{h \times w \times d}$ with spatial resolution $h \times w$ and channel dimension $d$. The binary masks $M_p$ and $M_g$ are downsampled into the latent space to obtain $M_p, M_g \in \{0,1\}^{h \times w \times 1}$.

To let the model to focus on the garment regions, we apply these masks to the image latents, yielding

$$
\begin{aligned}
z_p &= \varepsilon(I_p) \odot M_p, & z_b &= \varepsilon(I_p) \odot (1 - M_p), \\
z_g &= \varepsilon(I_g) \odot M_g, & z_c &= \varepsilon(I_c) \odot M_p.
\end{aligned}
\tag{4}
$$

Here, $z_p$ represents the person latent region to be updated with the given garment, $z_b$ corresponds to the background regions of the person latent to be preserved, $z_g$ captures the garment regions from the garment latent, and $z_c$ represents the garment regions from the pose latent.

At diffusion timestep $t$, noise is added to the person latent $z_p$ to produce $z_{p,t} = (1 - t)z_p + t\epsilon$, following Section 3. We then concatenate all latents and text tokens $c_{\text{text}}$ along the token dimension, forming $[z_{p,t}; z_b; z_g; z_c; c_{\text{text}}]$, and feed them into multiple multi-modal attention layers in DiTs. Note that this design fully leverage the inherent capability of DiTs to transfer garment appearance to the person, rather than relying on auxiliary encoders (Liu et al., 2024; Choi et al., 2024; Zhang et al., 2024).

To enable the model to distinguish spatially aligned conditions from non-aligned ones, inspired by (Tan et al., 2025a), we modify the RoPE embeddings such that $z_M$ and $z_c$, which are spatially aligned with $z_p$, share the same token positions as $z_p$, while $z_g$, which is not spatially aligned, is shifted by a fixed positional offset.

We apply an attention mask that restricts the conditioning latents to attend only to themselves, rather than interacting with the evolving person tokens $z_{p,t}$. This prevents the conditioning information from being corrupted during denoising and helps preserve fine garment details (Tan et al., 2025b).

We train this baseline under Equation 1, achieving competitive performance despite its simplicity. However, as shown in Figure 2, it still fails to transfer fine-grained garment details—logos and textures appear blurred, and structural boundaries lack sharpness. This motivates a closer analysis of the model's attention behavior, presented in the following section.

### 4.3 ANALYSIS

**Matching Cost from DiT Attention.** We further explore our baseline from the perspective of how the person and garment match each other within the DiT architecture. Leveraging the DiT's multi-modal attention capability, where person, garment, and text tokens interact, we specifically focus on the attention from the person latent $z_{p,t}$ to the garment condition $z_g$. This modifies Equation 2 and is formulated as

$$A^{l,h}_{z_{p,t}\to z_g} = \text{Softmax}\left(\frac{Q^{l,h}_{z_{p,t}}(K^{l,h}_{z_g})^\top}{\sqrt{d}}\right). \tag{5}$$

We redifine this $A^{l,h}_{z_{p,t}\to c_I}$ as the matching cost between the person and garment latents at timestep $t$. This is similar to the analysis in (Nam et al., 2024a; 2025a), but no prior work has framed the VTON task as person–garment correspondence within DiT attention.

**Correspondence Estimation.** We then extract person–garment correspondences from the matching cost $A^{l,h}_{z_{p,t}\to z_g}$ defined in Equation 5. We first average the matching costs across all heads to obtain $\bar{A}^l_{z_{p,t}\to z_g} \in \mathbb{R}^{N_q\times N_k}$, where $N_q$ and $N_k$ denote the numbers of person query tokens and garment key tokens, respectively. Each entry $\bar{A}^l_{z_{p,t}\to z_g}(i,j)$ represents the probability that the $i$-th person query token attends to the $j$-th garment key token. Finally, dense correspondences at layer $l$ are estimated by taking the $\arg\max$ over $j$:

$$\hat{X} = \{\hat{x}_i \mid i = 1, \ldots, N_q\}, \tag{6}$$

$$\hat{x}_i = \underset{j\in\{1,\ldots,N_k\}}{\arg\max} \ \bar{A}^l_{z_{p,t}\to z_g}(i,j). \tag{7}$$

**Pseudo-GT Construction.** Since there is no ground-truth correspondence between the person and garment images, we construct pseudo ground-truth matches using DINOv3, a strong vision foundation model for visual matching tasks (Siméoni et al., 2025). Given the person image $I_p$ and garment image $I_g$, we extract feature descriptors with $\phi(\cdot)$ from DINOv3. We then mask these descriptors with $m_c$ and $m_g$ to ensure that correspondences are computed only within the valid garment-related regions:

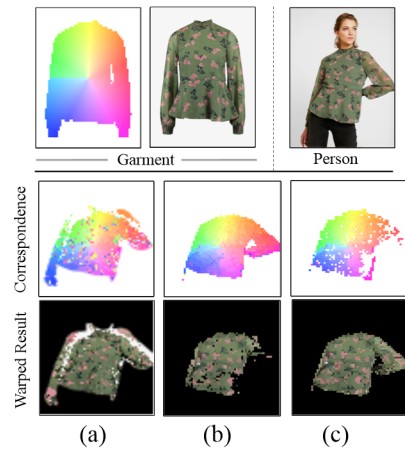

Figure 4: **Correspondence visualization.** (a) Baseline correspondences obtained from the multi-modal attention maps. (b) DINOv3 correspondences between the garment and person images. (c) Refined DINOv3 correspondences after applying the cycle-consistency check. The bottom row shows the garment warped to the person using each correspondence field.

$$\psi_p = \phi(I_p)\odot m_c, \quad \psi_g = \phi(I_g)\odot m_g. \tag{8}$$

With these masked descriptors, following the classic matching protocol (Hong et al., 2022a;b; Cho et al., 2024), we compute the cosine similarity as the matching cost:

$$C(i,j) = \frac{\psi_p(i)\cdot\psi_g(j)}{\|\psi_p(i)\|_2\,\|\psi_g(j)\|_2}, \tag{9}$$

where $i$ and $j$ index spatial locations of person and garment descriptors, respectively.

From this cost map, we calculate a person-to-garment flow, $F_{p\to g} \in \mathbb{R}^{h\times w\times 2}$ by taking the $\arg\max$ over garment locations, and similarly compute a garment-to-person flow $F_{g\to p} \in \mathbb{R}^{h\times w\times 2}$ by taking

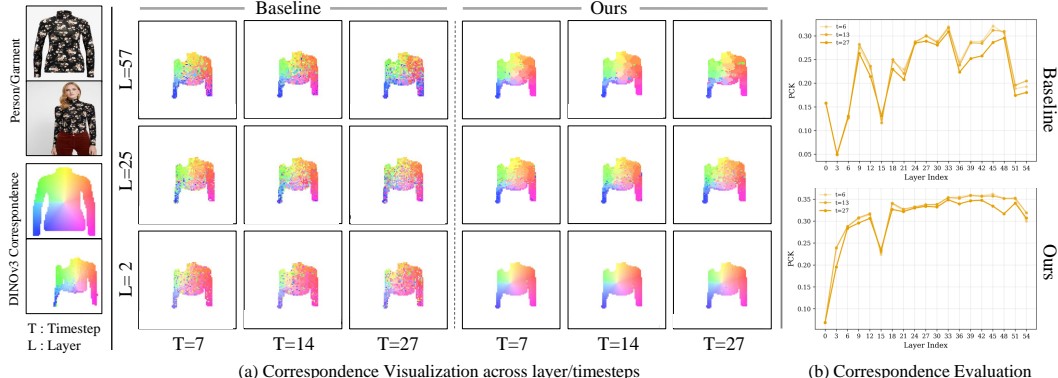

(a) Correspondence Visualization across layer/timesteps     (b) Correspondence Evaluation

Figure 5: **Visualization of attention-based correspondence estimation across timesteps and layers.** (a) *Top-left* shows the input garment and person images, and *bottom-left* shows the DINOv3 correspondences computed between the source garment and the target person. The grids on the right illustrate correspondences extracted from multi-modal attention maps, varying across timesteps (left to right) and layers (bottom to top). Overall, the baseline produces weak and unstable correspondences, while CORAL significantly improves the alignment with DINOv3, yielding more consistent person–garment matching. (b) We evauate whether predicted correspondences align well with DINOv3 by measuring percentage of coorect keypoints (PCK) on the VTION-HD (Choi et al., 2021) dataset.

the $\arg\max$ over person locations. To retain only reliable matches, we apply a cycle-consistency check defined as

$$\|F_{\mathrm{g}\to\mathrm{p}}(F_{\mathrm{p}\to\mathrm{g}}(i)) - i\| < \gamma, \tag{10}$$

where $\gamma$ is a small distance threshold. Only correspondences with cycle-consistency error below $\gamma$ are kept, as illustrated in Figure 4.

**Correspondence Evaluation.** Finally, we evaluate the predicted correspondences $\hat{x}_i$ by measuring the percentage of correct keypoints (PCK), the proportion of predictions within a distance $\alpha$ of the ground-truth correspondence $x_i^{\mathrm{GT}}$, which is defined as:

$$x_i^{\mathrm{GT}} = \underset{j \in \{1,\ldots,N_{\mathrm{k}}\}}{\arg\max} \ C(i, j). \tag{11}$$

Based on these correspondences, we measure alignment accuracy using the Percentage of Correct Keypoints (PCK):

$$\mathrm{PCK}(\alpha) \ = \ \frac{1}{N} \sum_{i=1}^{N} \mathbf{1}\left(\|\hat{x}_i - x_i^{\mathrm{GT}}\|_2 < \alpha\right). \tag{12}$$

Here, $N$ denotes the number of evaluated correspondences and $\alpha$ is a distance threshold.

**Observations.** In Figure 5(b), we observe that the correspondences estimated by the baseline exhibit significantly lower performance across all layers compared to those obtained from DINOv3. This indicates that the baseline struggles to establish accurate matches between person queries and garment keys, and suggests that this matching could be further improved by distilling pseudo ground-truth correspondences from DINOv3. Additionally, as shown in Figure 5(a) the baseline mostly captures left–right separation, while top–bottom structure clearly appears in only later layers. This indicates that the model only captures coarse structural layout in earlier layers, and does not fully establish complete garment alignment. Across timesteps, the overall patterns remain similar without clear improvement, suggesting that noise reduction alone does not resolve the correspondence problem. We also find that correspondences sharply degrade in later layers and timesteps. These observations show that correspondence errors occur throughout the denoising process, motivating us to apply supervision across all layers rather than restricting it to a few selected ones.

### 4.4 CORAL: CORRESPONDENCE ALIGNMENT FOR VIRTUAL TRY-ON

We propose CORAL, a lightweight alignment loss that can be seamlessly integrated into any DiT-based pipeline. Without modifying the underlying architecture, CORAL enhances person–garment matching by jointly optimizing two complementary supervision signals: (1) correspondence alignment loss and (2) entropy minimization loss.

**Correspondence Alignment Loss.** The goal of the correspondence alignment loss is to correct the query–key correspondences between the person and garment observed in the baseline by distilling reliable matches from DINOv3 (Siméoni et al., 2025). We focus on the attention map $A^{l,h}$ in Equation 3. After applying softmax, the attention maps are averaged across heads to obtain $\bar{A}^l$. The attended location for query $i$ is then computed through a soft-argmax, which interprets the attention weights as probabilities over key positions. Let $x_j \in \mathbb{R}^2$ denote the spatial coordinate of the $j$-th key token. The expected coordinate of query $i$ at layer $l$ is given by

$$\hat{x}_i^l = \sum_{j=1}^{N_{\mathrm{k}}} \bar{A}_{i,j}^l\, x_j, \tag{13}$$

Finally, we compute the averaged L2 loss between the estimated correspondences and the pseudo-GT matches, where $N_{\mathrm{c}}$ is the number of supervised query tokens inside the person mask $\mathrm{m}_c$ and $L$ is the total number of transformer layers:

$$\mathcal{L}_{\mathrm{corr}} = \frac{1}{LN_{\mathrm{c}}} \sum_{l=1}^{L} \sum_{i=1}^{N_{\mathrm{c}}} \|\hat{x}_i^l - x_i^{\mathrm{GT}}\|_2^2. \tag{14}$$

**Entropy Minimization Loss.** We introduce an entropy-based loss to explicitly supervise attention sharpness. Although the correspondence loss aligns queries with external matches, the alignment can still be unreliable when the attention distributions are too diffuse, as they may be dominated by weighted averages of incorrect key positions. Spatial entropy has been used to measure attention sharpness: Peruzzo et al. (Peruzzo et al., 2023) applied it as an inductive bias to promote localized attention in Vision Transformers, while Kang et al. (Kang et al., 2025) used it to identify heads focusing on text-relevant regions in vision–language models. Motivated by this, we employ entropy as a regularization objective to guide sharper query–key alignments. Following Shannon's definition of entropy (Attanasio et al., 2022), we consider the attention weights of a query as probabilities over keys and compute the entropy of this distribution. For a supervised query $i$ at layer $l$, the entropy of its attention distribution is

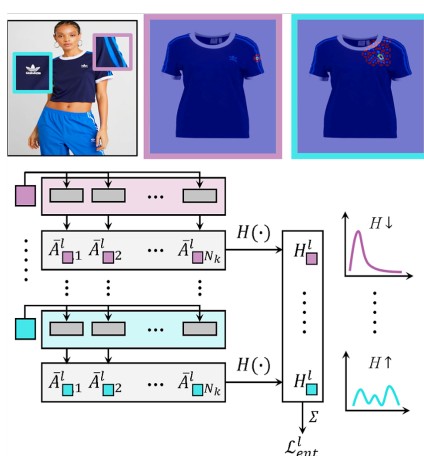

Figure 6: **Entropy minimization loss.** We compute the entropy of each query's attention distribution over keys and average across queries and layers.

$$H_i^l = -\sum_{j=1}^{N_{\mathrm{k}}} \bar{A}_{i,j}^l \, \log \bar{A}_{i,j}^l, \tag{15}$$

where $N_{\mathrm{k}}$ is the number of key tokens and $\bar{A}_{i,j}^l$ denotes the attention probability on key $j$. Averaging across queries and layers, the entropy loss is

$$\mathcal{L}_{\mathrm{ent}} = \frac{1}{LN_{\mathrm{c}}} \sum_{l=1}^{L} \sum_{i=1}^{N_{\mathrm{c}}} H_i^l. \tag{16}$$

Lower entropy corresponds to more localized matches, whereas higher entropy indicates diffuse or uncertain attention. This loss therefore complements the correspondence loss by enforcing sharper alignments across layers.

| Benchmark | Method | Paired | | | | Unpaired |
| | | FID ($\downarrow$) | KID ($\downarrow$) | SSIM ($\uparrow$) | LPIPS ($\downarrow$) | KID ($\downarrow$) |
|---|---|---|---|---|---|---|
| VITON-HD | Baseline | 9.52 | 1.35 | 0.864 | 0.084 | 0.456 |
| | Baseline + **CORAL** | 6.27 | 0.56 | 0.871 | 0.084 | 0.315 |
| DressCode | Baseline | 5.75 | 0.47 | 0.905 | 0.065 | 0.736 |
| | Baseline + **CORAL** | 5.22 | 0.56 | 0.905 | 0.060 | 0.193 |

Table 1: **Quantitative comparison.** We show results on VITON-HD and DressCode, demonstrating that our losses are effective in both paired and unpaired settings.

**Total Loss.** We extend the diffusion objective by adding correspondence and entropy terms:

$$\mathcal{L}_{\text{train}} = \mathcal{L}_{\text{diff}} + \lambda_{\text{corr}} \, \mathcal{L}_{\text{corr}} + \lambda_{\text{ent}} \, \mathcal{L}_{\text{ent}}, \tag{17}$$

where $\mathcal{L}_{\text{diff}}$ is explained in Eq 1. Instead of stage-wise training, we apply these objectives from the beginning, which gave better results. The weights $\lambda_{\text{corr}}$ and $\lambda_{\text{ent}}$ were set empirically.

## 5 EXPERIMENTS

### 5.1 EXPERIMENTAL SETTINGS

**Dataset** We choose to use two widely used public datasets, VITON-HD (Choi et al., 2021) and DressCode (Morelli et al., 2022) Models are trained separately on the training sets and evaluated on the corresponding test sets. VITON-HD includes 11,647 training pairs and 2,032 test pairs of upper-body garments with images at $1024 \times 768$ resolution. DressCode consists of 48,392 training pairs and 5,400 test pairs, covering upper-body, lower-body, and dress categories, also at $1024 \times 768$ resolution. Both datasets provide paired images of in-shop garments and person images.

**Metrics** Following prior works, we evaluate under two settings: *paired*, where the garment in the person image $I_p$ matches the garment image $I_g$, and *unpaired*, where the garments do not match and no ground-truth is available. For the paired setting, we use FID (Soloveitchik et al., 2022), KID Bińkowski et al. (2021), SSIM Wang et al. (2004), and LPIPS Zhang et al. (2018), while in the unpaired setting we use only FID and KID.

**Implementation Details.** Our model is based on `FLUX.1-dev` and fine-tuned with LoRA Hu et al. (2021). Training is conducted on two NVIDIA A100 GPUs for 7 epochs for VITON-HD and 3 epochs for DressCode with a batch size of 1. All results are generated at a resolution of $1024 \times 1024$. We use the Prodigy optimizer with a learning rate of 1.0. The correspondence and entropy objectives are weighted by $\lambda_{\text{corr}} = 10^{-3}$ and $\lambda_{\text{ent}} = 10^{-1}$.

### 5.2 EXPERIMENTAL RESULTS

**Quantitative Comparison.** We evaluate our method in comparison with the baseline on virtual try-on benchmarks VITON-HD (Choi et al., 2021) and DressCode (Morelli et al., 2022), with results shown in Tabs. 1. Compared to the baseline, our model achieves consistent gains: on VITON-HD we obtain FID decreases of 3.25 in the paired setup and KID decreases of 0.141 in the unpaired setup. On DressCode we also outperform the baseline across all garment types (upper body, lower body, dresses), with FID reductions of 0.53 in the paired setting and KID reductions of 0.543 in the unpaired setting.

**Qualitative Comparison.** Fig. 7 shows that our method achieves more accurate garment transfer compared to the baseline. First, fine details such as the exact color of small logos are well preserved, while the baseline often fails to reproduce these localized elements. Second, the baseline often attends to irrelevant regions of the garment and transfers incorrect parts. With our regularization, the model attends to the correct locations and transfers the proper regions, even in cases where visually similar areas can cause confusion.

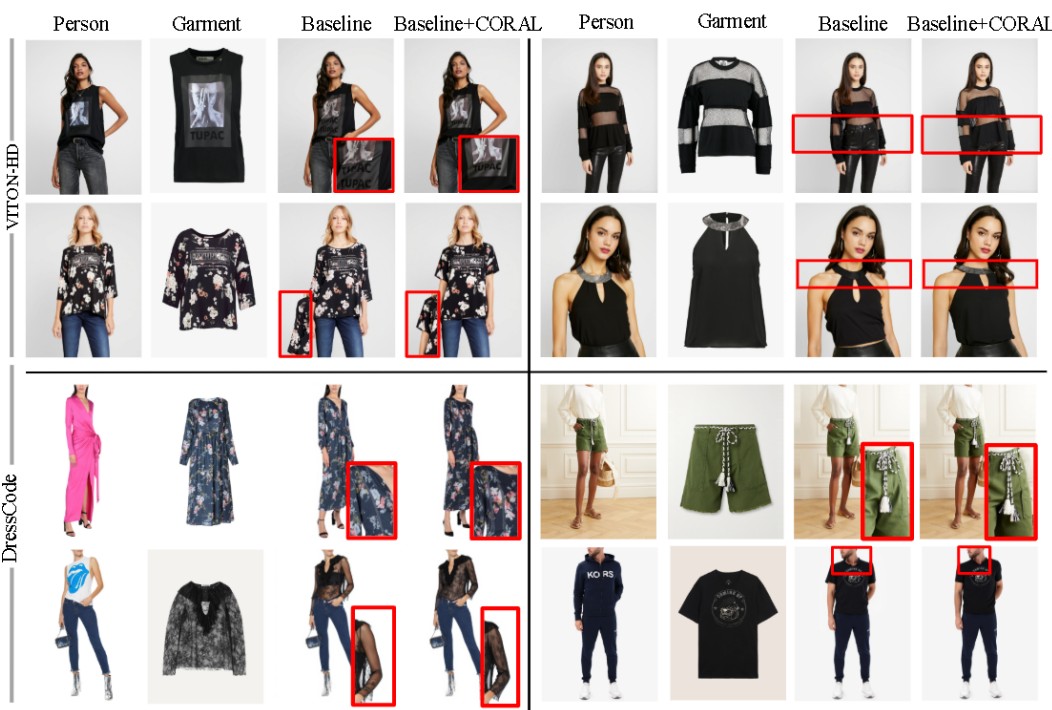

Figure 7: We compare our method with the DiT-based baseline under both paired and unpaired settings.

## 5.3 ABLATION STUDIES

**Impact of Different Losses.**    To analyze the contribution of each component, we perform ablations on the correspondence and entropy losses. Using $\mathcal{L}_{corr}$ alone improves correspondence quality and produces better alignment with correspondences from DINOv3, but entropy slightly increases compared to the baseline, showing that attention becomes more spread out even though correspondences are better aligned. Although details are improved , the model sometimes confuses garment length or fails on complex garment shapes. Conversely, applying only $\mathcal{L}_{ent}$ enhances attention sharpness and shows partial improvements, yet it does not sufficiently preserve fine-grained details or garment appearance. When both losses are applied together, the model achieves the best performance: correspondences are reliable, entropy is reduced, and garment details such as logos, textures, and shapes are preserved more consistently across diverse poses and clothing types.

## 6 CONCLUSION

We first systematically explore attention-based correspondence in DiT-based VTON models and observe that weak and diffuse query–key matching fundamentally limits fine-grained detail preservation. Motivated by these findings, we propose CORAL, which aligns person–garment correspondences with DINOv3 and enforces sharper attention distributions through entropy minimization. Experimental results on VITON-HD and DressCode demonstrate that CORAL consistently improves garment fidelity and structural alignment, offering a simple yet effective regularization strategy for future VTON pipelines.

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

## A  THE USE OF LARGE LANGUAGE MODELS (LLMS)

In accordance with the ICLR 2026 submission policy, we disclose that Large Language Models were used to assist in grammar correction, polishing of the writing in this paper and caption processing in our dataset curation pipeline.

