# OpenReview forum: "CORAL: Correspondence Alignment For Controllable Person Image Generation"
_ICLR.cc/2026/Conference — ICLR 2026 Conference Withdrawn Submission_

### Official Review · Reviewer_awEs · 2025-10-29

**Soundness:** 3
**Presentation:** 2
**Contribution:** 3
**Rating:** 4
**Confidence:** 5

**Summary:**

The paper proposes CORAL, a regularization strategy to improve person–garment alignment in diffusion-based virtual try-on. The core idea is to (1) align attention-based correspondences between person queries and garment keys using pseudo ground truth from DINOv3, and (2) sharpen attention with an entropy minimization loss so matches are more localized. The authors first analyze a DiT-based baseline by extracting dense correspondences directly from multi-modal attention and evaluating them with a PCK metric built from DINOv3 matches, finding weak and diffuse query–key alignment across layers/timesteps. CORAL then adds a correspondence loss (soft-argmax L2 to DINOv3 matches) and an entropy loss across layers. On VITON-HD and DressCode, CORAL improves FID/KID and preserves fine details like small logos and textures over a DiT baseline trained with standard diffusion losses. The method requires no architectural changes and is trained with LoRA on FLUX.1-dev.

**Strengths:**

1. Frames VTON as attention-level person–garment correspondence within DiTs, rather than only improving U-Net warping or photometric supervision. The DINOv3-distilled correspondence objective at attention level plus an explicit entropy regularizer is a neat combination that targets the failure mode (diffuse attention) head-on.
2. Presents a diagnostic framework: extracts attention correspondences across layers/timesteps and evaluates them against DINOv3-based pseudo-GT via PCK, revealing systematic weaknesses in the baseline and motivating the loss design. Figures show layer/timestep grids and quantitative PCK trends.
3. The losses are clearly specified (soft-argmax coordinate expectation; Shannon entropy over attention; total loss with λ’s), and training/inference details (datasets, metrics, LoRA/FLUX.1-dev setup) are provided.
4. Empirical gains over the baseline are consistent on two datasets across paired/unpaired settings (FID/KID, with SSIM/LPIPS in paired).
5. If attention alignment is indeed a key bottleneck in diffusion VTON, a plug-and-play regularizer that materially sharpens and aligns attention could be widely adopted across DiT-style pipelines and potentially beyond..

**Weaknesses:**

1. Missing SOTA comparison table. The quantitative table compares only Baseline vs Baseline+CORAL, with no head-to-head against contemporary VTON systems (e.g., StableVITON, IDM-VTON, Leffa, CATVTON, etc.). For ICLR, a comprehensive table across standard metrics/datasets is expected. This omission makes it hard to judge real-world significance beyond the authors’ baseline. (See Table 1: only two rows per dataset.)
2. Figure polish and presentation quality. Many figures (attention maps, correspondence grids, qualitative visuals) look rough and under-annotated: axis labels, scale bars, consistent color maps, and high-resolution crops are needed for ICLR standards. Some composites are cramped; captions could better specify layers/timesteps/heads and exact visualization procedures. (E.g., multi-panel correspondence visualizations and qualitative comparisons.)
3. Ablation/detail gaps. Loss-weight sensitivity (λ_corr, λ_ent) is not explored; the text notes empirical choices but no curves or robustness ranges.
4. Overhead analysis is missing: how much training cost (DINOv3 feature extraction, added losses) and inference cost (if any) does CORAL add? The analysis reports layer/timestep behavior, but a systematic entropy profile (before/after, per layer) and PCK@α curves would strengthen the case.
5. Only VITON-HD and DressCode are reported; no in-the-wild or cross-domain robustness tests.

**Questions:**

1. Please add a full comparison table on VITON-HD and DressCode versus recent methods cited in your related work, reporting FID/KID and standard paired metrics (SSIM/LPIPS). If possible, include per-category (upper/lower/dress) breakdowns on DressCode.
2. Overhead and practicality. What is the training compute/time overhead of CORAL (DINOv3 feature extraction + extra losses) relative to the baseline? Is there any inference-time cost? A table with wall-clock and GPU memory would clarify practicality.
3. Loss behavior and stability. Provide λ_corr/λ_ent sensitivity (e.g., grid or sweep) and report how attention entropy and PCK change.
Do you observe failure modes (e.g., over-sharp/peaky attention that latches onto wrong keys)? Any mitigation (temperature scaling, head dropout)?
4. Analysis depth. Please include layer-wise entropy curves (before vs after) and PCK@α curves to complement the qualitative grids.
Can you show failure cases (repetitive patterns, text, specular materials) with attention overlays and discuss why CORAL fails/succeeds?
5. Suggestions for rebuttal. Substantially upgrade figure quality (resolution, consistent layouts, readable colormaps/legends, clearer captions; include high-res zooms on logos/patterns and side-by-side with SOTA).

---

### Official Review · Reviewer_soKg · 2025-11-01

**Soundness:** 2
**Presentation:** 3
**Contribution:** 2
**Rating:** 4
**Confidence:** 4

**Summary:**

This paper addresses the virtual try-on task, which aims to synthesize a person wearing a specific garment, given an identity image of the person and an image of the target clothing. The authors specifically focus on establishing the correspondence between the human body and the clothes. To this end, they propose aligning the attention map of the generation model with a pseudo ground truth derived using DINOv3. Furthermore, they introduce an attention entropy loss to regularize the sharpness of the attention map, thereby encouraging it to focus more precisely on the corresponding region.

**Strengths:**

The rationale for this work is clear and well-justified.

The paper is clearly presented and well-structured.

The empirical results demonstrate a strong performance.

**Weaknesses:**

1. Lack of sufficient comparison with previous virtual try-on methods. All results appear to be conducted against a simple baseline. Since many other virtual try-on methods exist, a broader comparative evaluation is necessary.

2. Lack of sufficient discussion regarding previous correspondence alignment methods. Papers such as [1] and [2] also discuss the utilization of correspondence. The authors should provide a fair comparison or discussion of how their method relates to these existing approaches.

[1] Wear-Any-Way: Manipulable Virtual Try-on via Sparse Correspondence Alignment. ECCV 2024.

 [2] INCORPORATING VISUAL CORRESPONDENCE INTO DIFFUSION MODEL FOR VIRTUAL TRY-ON. ICLR 2025.

**Questions:**

1. The discussion of related work and previous methods is insufficient. (Please refer to the **Weakness** section.)
2. Figure 4 is counter-intuitive and requires clarification: Please explain why the seemingly suboptimal correspondence in panel (c) (compares with (b)) leads to superior final wrapped results.

---

### Official Review · Reviewer_D2C4 · 2025-11-01

**Soundness:** 2
**Presentation:** 3
**Contribution:** 2
**Rating:** 2
**Confidence:** 4

**Summary:**

This paper focuses on the research of Virtual Try-On (VTON) in the field of controllable person image generation. It identifies a critical limitation of existing diffusion model-based VTON methods: their inability to preserve fine-grained clothing details (e.g., logos, patterns, textures). The root cause of this issue is pinpointed as inaccurate query-key matching in attention maps. To address this problem, the paper makes two core contributions: first, it proposes a "correspondence evaluation framework" that extracts dense correspondences from attention maps and conducts evaluation using pseudo-ground-truth matching; second, it introduces the CORAL lightweight regularization strategy, which consists of a "correspondence loss" (aligning query attention with reliable external matches) and an "entropy loss" (sharpening attention to improve matching confidence). Notably, the CORAL strategy can be adapted to existing diffusion models without modifying their architectures.

**Strengths:**

The proposed methods exhibit strong logical coherence: the evaluation framework identifies attention misalignment, while the CORAL strategy directly addresses this issue through targeted loss functions. This end-to-end problem-solution pipeline ensures the methods are theoretically sound and well-aligned with the research goal.

**Weaknesses:**

The core idea of "correspondence learning" in this paper bears notable similarities to the method proposed in Cross-domain Correspondence Learning for Exemplar-based Image Translation. However, the paper fails to explicitly highlight the essential differences between the two works.

This paper heavily lacks comparison with state-of-the-art works in virtual try-on field

**Questions:**

No detailed ablation study are provided？

---

### Note · Authors · 2025-11-12

I have read and agree with the venue's withdrawal policy on behalf of myself and my co-authors.